# Effects of Process Parameters on Bond Properties of Ag-2.35Au-0.7Pd-0.2Pt-0.1Cu Alloy Wire

**DOI:** 10.3390/mi14081587

**Published:** 2023-08-12

**Authors:** Hongliang Zhou, Andong Chang, Junling Fan, Jun Cao, Yingchong Zhang, Bin An, Jie Xia

**Affiliations:** 1School of Mechanical and Power Engineering, Henan Polytechnic University, Jiaozuo 454000, China; cadzmy@163.com (A.C.); a1951198730@163.com (B.A.); x6868mrtx@163.com (J.X.); 2School of Chemical and Environmental Engineering, Jiaozuo University, Jiaozuo 454000, China; jzufanjunling@163.com; 3Nanjing High Speed Gear Manufacturing Co., Ltd., Nanjing 211100, China; 212005010007@home.hpu.edu.cn

**Keywords:** bond properties, EFO current, EFO time, ultrasonic power, bonding force

## Abstract

Bond properties were performed on Ag-2.35Au-0.7Pd-0.2Pt-0.1Cu alloy wire with a diameter of 25 µm under different process parameters. The effects of electrical flaming off (EFO) current and EFO time on the deformability of the free air ball (FAB) were investigated using scanning electron microscopy (SEM), as well as the effects of ultrasonic power and bonding force on the bond characteristic. The experimental results show that FAB grows from a preheated tip to a small ball, a regular ball, and finally to a golf ball with increasing either the EFO current or the EFO time, and the FAB presents an optimal shape at 25 mA and 650 μs. Moreover, a nonlinear relationship between FAB diameter and EFO time is obtained at an EFO current of 25 mA, which could be expressed by a cubic equation. Further, at a constant bonding force, as the ultrasonic power increased, the mashed ball diameter grew larger and larger, the capillary hole imprint became more and more obvious, and the tail width also increased, and vice versa. The optimal ultrasonic power and bonding force are 70 mW and 45 gf for ball bonding and 90 mW and 75 gf for wedge bonding, respectively. Finally, for all the bonded wire samples prepared under optimal process parameters, no ball and wedge bond lifts happened after the destructive pull test, and full intermetallic compound coverage with perfect morphology occurred on the bond pad after the ball shear test, which meant that the bonded wire samples had high bond strength and hence improved the reliability of microelectronic products. It provided technical support for the reliability research of Pt-containing Ag-based bonding alloy wires.

## 1. Introduction

Microelectronics packaging is an overarching part of the microelectronics industry, which is one of the main pillar industries in the world. Microelectronic products have been extensively used in all fields of social life and are driving the continuous development and technological innovation of the overall electronic industry, whose quality and lifespan are influenced by microelectronic packaging technology [1]. Wire bonding is the most common technique for forming electrical interconnections in the microelectronics industry because of its cost-effectiveness, flexibility, and robustness. It forms the primary semiconductor packages in the field of microelectronic packaging currently [2,3,4]. The bonding wire, an essential structural material for microelectronic packaging, plays a significant role in connecting the integrated circuit (IC) chips to the metal lead frame [5].

Au bonding wire, as the earliest and most widely used bonding wire, has excellent mechanical and electrical properties, high reliability, and ease of assembly. However, due to the increasingly high cost of Au and its limited performance development in recent years, alternative bonding wires have been considered [6,7,8,9]. Cu bonding wire is a preferred alternative material because of its lower cost, slower intermetallic growth on Al pads, and higher thermal conductivity and mechanical strength compared with Au bonding wire [10,11,12]. However, researchers agree that Cu bonding wire is not the definitive substitute for Au bonding wire due to its high hardness and oxidation rate and complex bonding process that will result in pad damage during bonding [13,14,15]. Ag bonding wire, as a promising material for the development trend of microelectronic packaging with high density, high integration, and high speed, has lower cost, better manufacturability, and higher reliability than Au and Cu bonding wires [16,17,18], and it has been utilized in integrated circuit (IC) and light-emitting diode (LED) packaging [19,20,21].

However, pure Ag wire has some problems such as easy oxidation, easy collapse, and Ag+ migration that make it inconvenient to apply. Alloying can solve the above problems and provide an effective way to develop new Ag bonding wires of low cost and high reliability [22]. Fan et al. [23] studied the bond strength and reliability of Ag-Au alloy wires with different Au contents (1 wt%, 3 wt%, and 5 wt%) and found that with the increase in Au content, the size of the heat-affected zone (HAZ) decreases, the tension and shearing force of the free air ball (FAB) increase, and when the Au content is 5 wt%, the best FAB morphology can be obtained. Ag alloy wires with various contents of Pd and Au were evaluated by Tsai et al. [24]. The experimental results presented showed that their breaking load increases with Au and Pd contents. Yuan et al. [25] compared Ag alloy wires with different Pd contents (3 wt%, 4 wt%, and 6 wt%) and pure Ag wires and found that the addition of Pd improved the mechanical properties of Ag wires, and as the Pd content increased, the bonding strength of Ag alloy wires increased, and adding Au to the alloy wire in the Ag wire with added Pd will improve the thermal stability of the Ag alloy wire. Chuang et al. [26,27] provided that an innovative Ag-8Au-3Pd alloy wire with a high twin density can be produced via appropriate drawing and annealing processes that exhibit high thermal stability during high temperature exposure.

The above-described studies are mostly performed on the performance of Ag alloy wires with Au and Pd elements, in the present study, a new type of Ag alloy wire, named Ag-2.35Au-0.7Pd-0.2Pt-0.1Cu alloy (AAPPCA) wire, is obtained by further adding Pt and Cu elements to triple Ag-Au-Pd alloy. Pt has excellent chemical stability, and adding Pt can shorten the length of the heat-affected zone and reduce the diameter of the FAB [28]. The FAB morphology and bonding strength of APPAC wire were studied, and then the optimal electrical flaming off (EFO) and bonding parameters for the AAPPCA wire were obtained. This study obtained a new type of Ag-based bonding alloy wire by adding Pt element and obtained suitable firing balls and bonding parameters, providing a new approach for the preparation of other types of Ag-based bonding wires. Hopefully, the results of this study could serve as the academic and theoretical basis for the use of AAPPCA wire in microelectronic packaging.

## 2. Materials and Methods

### 2.1. Experimental Equipment

A KAIJO FB-988 automatic bonding machine made in Japan was employed in this study. The packaging type 2835 LED and capillary type SPT SU-50140-425E-ZU36-E were used, and an AAPPCA wire with a diameter of 25 µm was also employed here, whose average elongation and tensile strength are 11.5% and 259.1 MPa, respectively, as shown in Figure 1. N_2_ with a flow rate of 0.6 L/min was used as a shielding gas to prevent the oxidation of the molten AAPPCA wire during the EFO process, and the process parameters are shown in Table 1. The morphology of FAB and bonds was observed using a scanning electron microscope (SEM) system, and the relationship between FAB diameter and EFO time was fitted. The bond strength was measured using a DAGE Series 4000 BS250 system at a test speed of 500 µm/s and a shear height of 5 µm.

### 2.2. Test Method for Bonding Strength

#### 2.2.1. Destructive Pull Test

The destructive pull test is the most widely used technique to evaluate the bond strength by hooking and applying a normal upward force and pulling the bonded wire until it fails, and the major failure modes that one observes during a bond pull test are ball bond lift (Location A), ball neck break (Location B), midspan wire break (Location C), heel break (Location D), and wedge bond lift (Location E), as shown in Figure 2. The test results are important evidence for certifying the proper setup of the process parameters and evaluating bonding quality and reliability. A wire break at B, C, or D is the preferred break during the pull strength test, which indicates a strong bond between the wire and the metallization.

#### 2.2.2. Ball Shear Test

The destructive pull test is suitable for most applications. However, it provides very little information on the ball bond strength and quality, so it fails to determine the true ball bond strength. This factor has led to the development of the ball shear test, which is performed using a shear tool to push off the ball bond with sufficient force. During the shear testing process, the shear ram is located directly above the bond pad, and the bottom of the shear tool is close to the center of the ball. The shear tool moves parallel to the bonding surface and shears the ball bonding. The principle of the ball shear test is shown in Figure 3, whose results can reflect the intermetallic formation and its coverage of ball bonds.

## 3. Results and Discussion

### 3.1. Effects of EFO Current and EFO Time on FAB Morphology of AAPPCA Wire

During the EFO process, inappropriate EFO parameters will result in FAB with imperfect morphology and sphericity [29] and then obviously cause poor ball bonding contact, which has a bad effect on the bonding yield [30]. It is closely related to the solidification process during FAB formation, so optimal EFO parameters should be chosen carefully.

Figure 4 shows the FAB morphologies of AAPPCA wire for different EFO times at a constant EFO current of 20 mA. It could be seen that the wire tip is preheated by the EFO discharge from 550 μs to 600 μs, as shown in Figure 4a,b. When the EFO time increases from 550 μs to 750 μs, the bonding wire has a small FAB with poor sphericity since the EFO current of 20 mA generates inadequate energy to melt the wire tail, which results in a smaller FAB than a standard one even under the effect of surface tension. What is worse, a hollow is found at the bottom of FAB due to a lack of energy input via arc discharging, as shown in Figure 4c–e, which is unacceptable in practical applications. It indicated that a higher EFO current than 20 mA is needed to form a standard FAB of the AAPPCA wire to satisfy the bonding yield.

Figure 5 shows the FAB morphologies of AAPPCA wire for different EFO times at a constant EFO current of 25 Ma. It was found that the FAB diameter at 25 Ma is larger than that at 20 Ma for the reason that a higher EFO current of 25 mA generated enough energy to melt the wire tail completely. At 550-650 μs, a regular FAB with a smooth surface solidifies from its neck to its bottom until it achieves the equilibrium position, as shown in Figure 5a–c. While at 700–750 μs, the off-center FAB can be observed from the AAPPCA wire, as shown in Figure 5d,e. A possible reason was that the spark from the EFO electrode reaches the wire on the side, which makes the wire tail experience a lower surface temperature on the opposite side, and thus the molten wire will lean to the spark side due to unbalanced surface tension on the two sides. Finally, a golf-clubbed FAB occurs after natural cooling, which will result in misplacement of the bond, weak bonding, or bond lift and therefore worsen the bond’s strength and reliability.

When the EFO current turned to 30 mA, the FAB formed rapidly because of the great amount of heat due to the large current, which caused the FAB center to deviate from that of the AAPPCA wire. As a result, more golf-clubbed FAB comes out at 550–750 μs, as shown in Figure 6. In consequence, the short circuits probably occur in fine-pitch ball bonding, which cannot be accepted.

Considering the occurrence of off-center FAB at the EFO time of 25 mA and 700 μs, ten FAB samples forming at 25 mA and 650 μs were chosen to make further morphology observations using the SEM system. The results show that all ten FAB samples have a smooth surface without defects and an approximate diameter consistent with perfect sphericity. The average diameter of the ten FAB samples was about 45.5 µm, as shown in Figure 7, which was 1.82 times that of the AAPPCA wire and in line with production requirements. In conclusion, the optimal EFO current and time are 25 mA and 650 μs, respectively.

### 3.2. Relationship between FAB Diameter of AAPPCA Wire and EFO Time

During the dynamic FAB formation process, EFO time is a key factor that dominates the FAB diameter at a constant EFO current, which affects the FAB growth rate and model. Generally, a FAB with a small diameter will directly lead to a decrease in bonding strength, while a FAB with a large diameter will reduce bonding density [28].

The relationship between the FAB diameter and EFO time of AAPPCA wire at the EFO current of 25 mA was obtained via experiment, as shown in Figure 8, from which it could be seen that the FAB diameter increases relatively quickly from 500 μs to 750 μs, while slowly after 750 μs. A cubic equation representing the relationship between the FAB diameter and EFO time was obtained by fitting the experimental data using the least squares method.
(1)$$d=-9.33327\times 10^{-7}t^{3}+0.00185t^{2}-1.16416t+276.04126$$

*d* is the FAB diameter; *t* is the EFO time.

Further, the correlation coefficient, r, of 0.989 was also obtained by the least squares method with a confidence interval of 500–800 μs. It means that the above cubic equation can be used by the microelectronics industry to determine the EFO time for AAPPCA wire with a diameter of 25 μm.

### 3.3. Effects of Ultrasonic Power and Bonding Force on Ball Bond Morphology of AAPPCA Wire

Ultrasonic power and bonding force are key factors affecting bondability in the bonding process. The contamination and oxide layer can be broken down by ultrasonic vibration to improve the interfacial adhesion between FAB and pad metal. Ultrasound energy can lower the bonding force by softening FAB effectively and then increasing the dislocation density, thereby lowering the flow stress. The bonding force produces sliding friction at the interface, which leads to some wear and then to the bond being formed.

Figure 9 shows the SEM images of ball bands for various ultrasonic powers, ranging from 60 mW to 80 mW, at a constant bonding force of 35 gf. The mashed balls in Figure 9a–c had relatively small diameters of 54.7 μm, 55.2 μm, and 56.4 μm, respectively, due to the small bonding force of 35 gf, which would lead to low bond strength. What was worse, off-center ball bonds were also found in these cases, as shown in Figure 10, and thus had bad effects on bond quality.

Figure 11 shows the ball band morphologies for various ultrasonic powers, ranging from 60 mW to 80 mW, at a constant bonding force of 45 gf. The mashed ball diameters in Figure 11a–c are 57.6 μm, 59.4 μm, and 59.7 μm, respectively, all of which were larger than that in Figure 10a–c. It means that a larger bonding force can produce a larger contact area between FAB and pad metal and therefore improve bond strength. However, the oval mashed ball occurred at 80 mW, whose diameter only increased by 0.3 μm compared with that at 70 mW; it would cause poor bond strength.

When the bonding force was 55 gf, the ball band morphologies for various ultrasonic powers, ranging from 60 mW to 80 mW, were obtained and are shown in Figure 12a–c. It could be seen from Figure 12a that Al splash occurs at 60 mW because of the large bonding force of 55 gf, which would increase the risk of short circuit or circuit function disorder. Especially for the ball bonds shown in Figure 12b,c, which formed at the bond pad edge, a large bonding force might cause cracking of the passivation at the bond pad edge or might even lead to cracking of the substrate. Hence, the optimal ultrasonic power of 70 mW and bonding force of 45 gf were chosen for the ball bond of AAPPCA wire in order to ensure ball bond reliability.

### 3.4. Effects of Ultrasonic Power and Bonding Force on Wedge Bond Morphology of AAPPCA Wire

The wedge bond, also called a stitch bond, is formed by the application of bonding force and ultrasonic energy by deformation of the wire between the capillary and the substrate, or lead finger. Tail formation is in the last step in the bonding process as well as a necessary step to continue the bonding process. Stable wedge bond formation plays a significant role in the stability of the thermosonic wire bonding process.

Figure 13 shows the wedge band morphologies for various ultrasonic powers, ranging from 80 mW to 100 mW, at a constant bonding force of 55 gf. It could be found that the tail width increases with increasing ultrasonic power. In Figure 13a, the tail bond detached away from the substrate due to the inadequate ultrasonic power at 55 gf, which represented bad bond quality and maximized production stoppages. No capillary hole imprint (CHI) in Figure 13b and an indistinct one in Figure 13c both indicated poor stitch bondability between wire and substrate and posed a risk to the long-term reliability.

Figure 14 shows the wedge band morphologies for various ultrasonic powers, ranging from 80 mW to 100 mW, at a constant bonding force of 75 gf. Compared with Figure 14a, Figure 15b exhibits more than half of the CHI circle circumference and a more symmetrical tail shape, which indicated that the wedge bond in Figure 14b has a higher bond strength than that in Figure 14a. While Figure 14c shows complete CHI and a asymmetric tail, the wedge bond increased the risk of substrate damage and bond failure.

Figure 15 exhibits the wedge band morphologies for various ultrasonic powers, ranging from 80 mW to 100 mW, at a constant bonding force of 95 gf. In such a case, complete CHI could be found due to the large bonding force, which became more and more obvious with increasing ultrasonic power. What was worse, more asymmetric tails occurred in such case, as shown in Figure 15b,c, and the tail width at 95 gf was larger than those at 55 gf and 75 gf, since the larger bonding force at the wedge could squeeze enough material at the heel of the bond, resulting in a bond failure easily, which should not be accepted. So the optimal ultrasonic power of 90 mW and bonding force of 75 gf were chosen here for the wedge bond of AAPPCA wire in order to ensure wedge bond quality.

### 3.5. Bond Strength Test Results

Here, twenty bonded wire samples, obtained at optimal process parameters (25 mA and 650 μs for FAB, 70 mW and 45 gf for ball bond, 90 mW and 75 gf for wedge bond), were chosen randomly to carry on the destructive pull test, and the test results were shown in Table 2, it was obvious that all the bonded wire samples break at B, C or D, no failure happened at A and D. Moreover, it could be clearly seen from Figure 16 that ball neck break is the predominant failure mode for bond wire samples; a possible reason was that there was the highest stress concentration at B, which would easily induce ball neck cracking failure. The pull force ranged from 9.6 gf to 13.4 gf, as shown in Figure 17, and the minimum value of the pull force is larger than the standard one [31], which means that all bonded wire samples have high bond strength.

The shear force results of the twenty bonded wire samples are shown in Table 2 and range from 33.2 gf to 45.7 gf, as shown in Figure 16. The minimum value of shear force is also larger than the standard one [31]. Figure 18 presents a typical SEM image of ball band after being sheared. Full intermetallic compound (IMC) coverage with regular morphology was found on the bond pad, which enhanced the adhesion of bond to pad. The above pull and shear test results show that the bonded wire samples have enough bond strength and hence improve the reliability of microelectronic products.

## 4. Conclusions

Our research has yielded the following conclusions:(1)A new type of AAPPCA wire was proposed that has high bonding strength and reliability.(2)As the EFO time increases from 550 μs to 750 μs, the FAB of AAPPCA wire grows from a preheated tip to a small ball with a hollow at its bottom at 20 mA, while it changes from a small ball to a regular ball and finally to a golf ball at 25 mA. When the EFO current is 30 mA, all the FABs exhibit golf balls. The EFO current and time for a regular and smooth FAB are 25 mA and 650 μs, respectively.(3)For the AAPPCA wire, at a constant EFO current of 25 mA, the relationship between the FAB diameter and EFO time can be expressed by a cubic equation obtained by fitting the experimental data using the least squares method.(4)For ball bonding, the mashed ball diameter of AAPPCA Wire increases with increasing either the ultrasonic power or the bonding force. For wedge bonds, with the increase in ultrasonic power or bonding force, the CHI becomes more and more obvious, and the tail width increases larger and larger. The optimal ultrasonic power and bonding force are 70 mW and 45 gf for ball bonding and 90 mW and 75 gf for wedge bonding, respectively.(5)The destructive pull test results show that all the bonded wire samples obtained at optimal process parameters break at B, C, or D, and full IMC coverage with regular morphology occurs on the bond pad after the ball shear test. So, all the bonded wire samples have enough bond strength, which is beneficial to the reliability of microelectronic products. It provides technical support for the research of Pt-containing Ag-based alloy wire.

## Figures and Tables

**Figure 1 micromachines-14-01587-f001:**
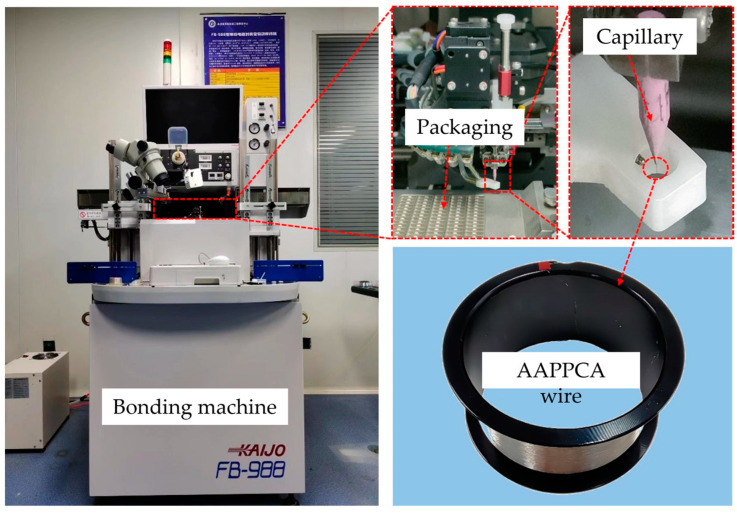
Experimental equipment and materials.

**Figure 2 micromachines-14-01587-f002:**
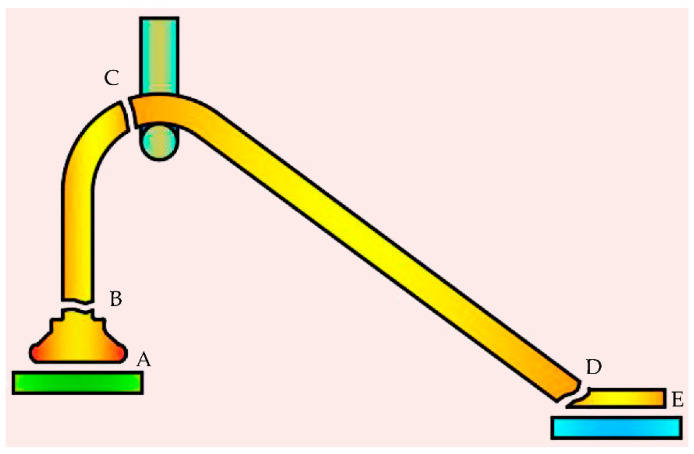
Pull test failure modes. Main failure modes and locations: A. Ball bond lift; B. ball neck break; C. midspan wire break; D. heel break and E. wedge bond lift.

**Figure 3 micromachines-14-01587-f003:**
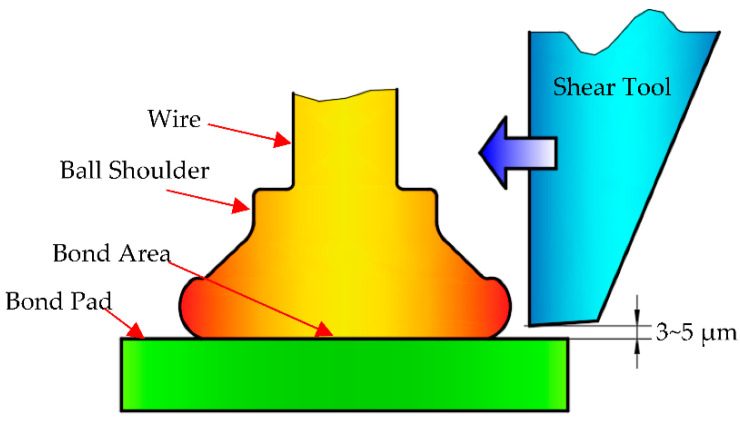
Ball shear test.

**Figure 4 micromachines-14-01587-f004:**
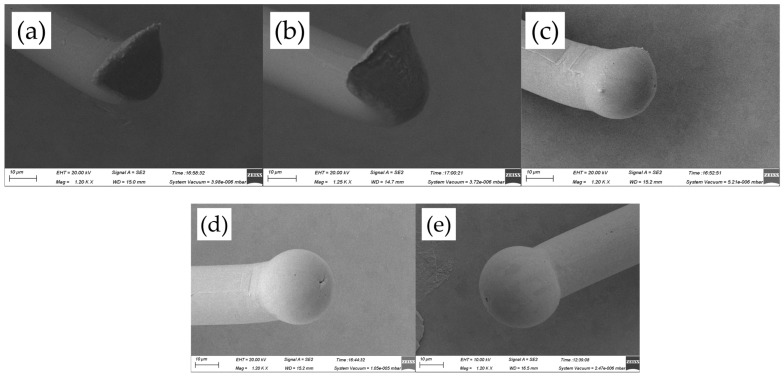
FAB morphologies of AAPPCA wire for different EFO times at 20 mA: (**a**) 550 μs, (**b**) 600 μs, (**c**) 650 μs, (**d**) 700 μs, (**e**) 750 μs.

**Figure 5 micromachines-14-01587-f005:**
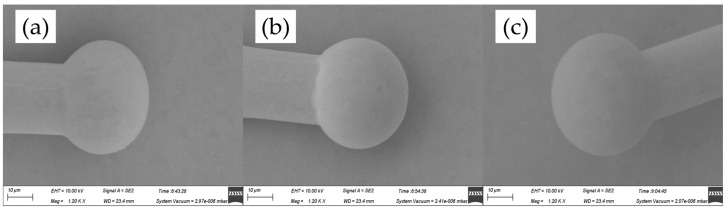
FAB morphologies of AAPPCA wire for different EFO times at 25 mA: (**a**) 550 μs, (**b**) 600 μs, (**c**) 650 μs, (**d**) 700 μs, (**e**) 750 μs.

**Figure 6 micromachines-14-01587-f006:**
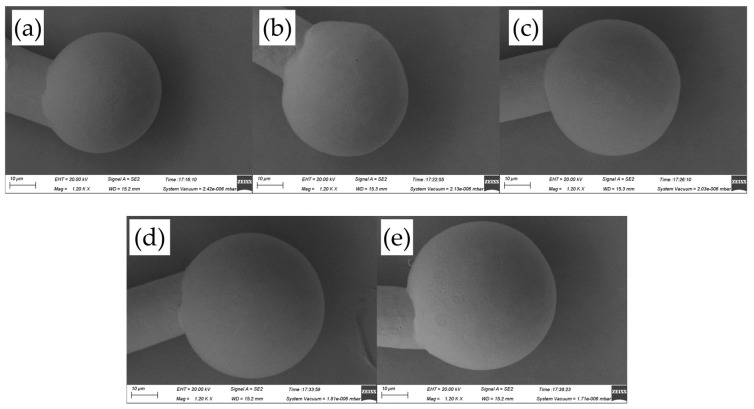
FAB morphologies of AAPPCA wire for different EFO times at 30 mA: (**a**) 550 μs, (**b**) 600 μs, (**c**) 650 μs, (d) 700 μs, (**e**) 750 μs.

**Figure 7 micromachines-14-01587-f007:**
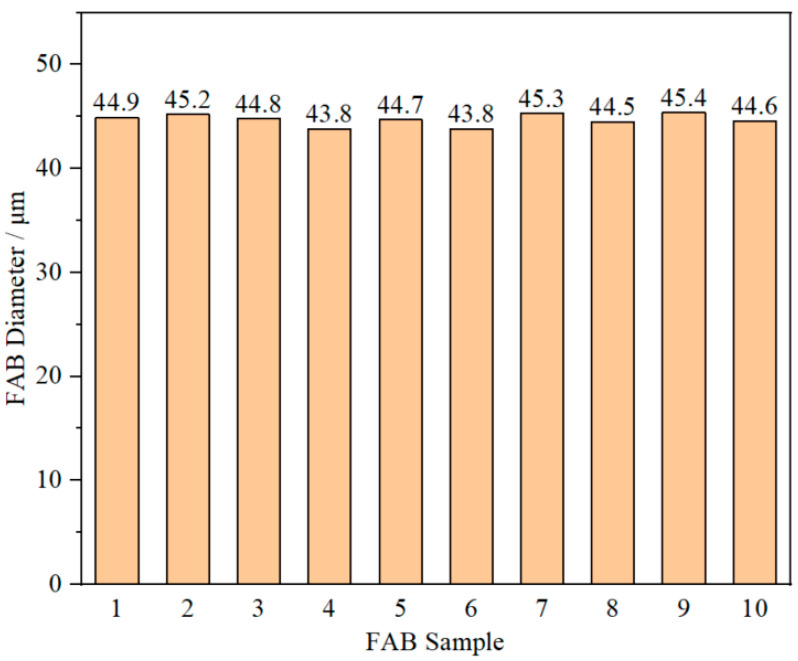
FAB diameter of the ten FAB samples forming at 25 mA, 650 μs.

**Figure 8 micromachines-14-01587-f008:**
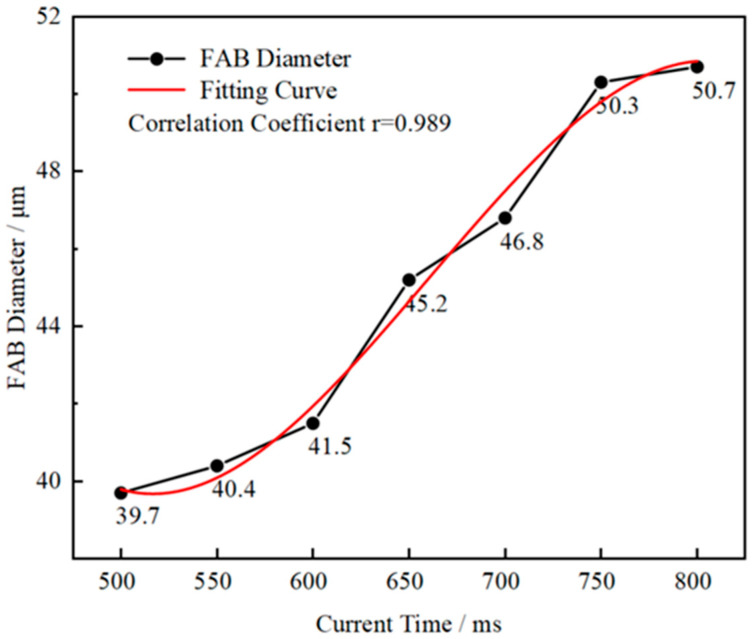
Relationship between the FAB diameter and EFO time of AAPPCA wire at the EFO current of 25 mA.

**Figure 9 micromachines-14-01587-f009:**
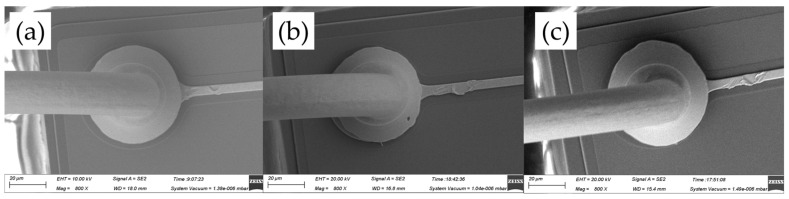
Ball bond morphologies of AAPPCA wire for different ultrasonic powers at 35 gf: (**a**) 60 mW, (**b**) 70 mW, (**c**) 80 mW.

**Figure 10 micromachines-14-01587-f010:**
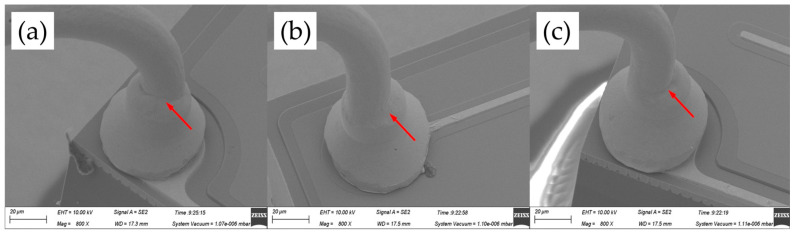
Off-center ball bond morphologies of AAPPCA wire for different ultrasonic powers at 0.35 N: (**a**) 60 mW, (**b**) 70 mW, (**c**) 80 mW. The arrow indicates off-center ball bonds.

**Figure 11 micromachines-14-01587-f011:**
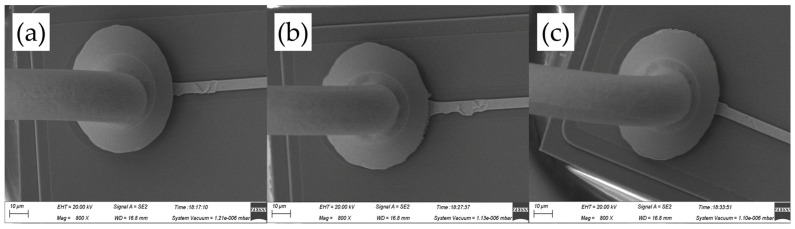
Ball bond morphologies of AAPPCA wire for different ultrasonic powers at 45 gf: (**a**) 60 mW, (**b**) 70 mW, (**c**) 80 mW.

**Figure 12 micromachines-14-01587-f012:**
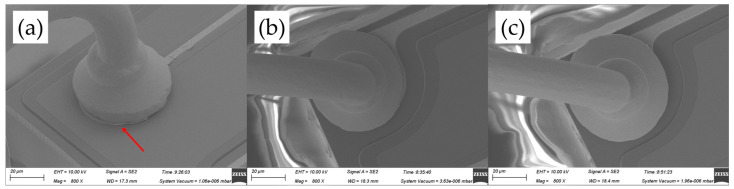
Ball bond morphologies of AAPPCA wire for different ultrasonic powers at 55 gf: (**a**) 60 mW, (**b**) 70 mW, (**c**) 80 mW. The arrow indicates large bonding force.

**Figure 13 micromachines-14-01587-f013:**
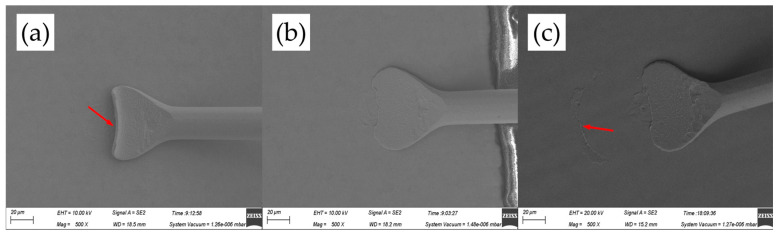
Wedge bond morphologies of AAPPCA wire for different ultrasonic powers at 55 gf: (**a**) 80 mW, (**b**) 90 mW, (**c**) 100 mW. The arrow in Figure 13a points to the tail bond detached away from the substrate. The arrow in Figure 13c points to an indistinct imprint.

**Figure 14 micromachines-14-01587-f014:**
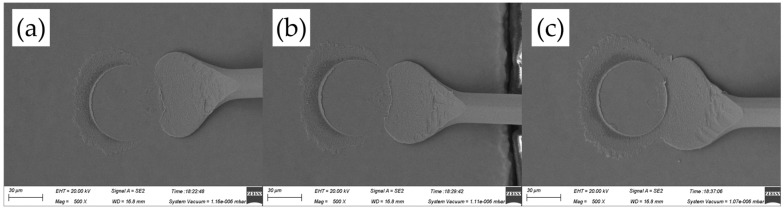
Wedge bond morphologies of AAPPCA wire for different ultrasonic power at 75 gf: (**a**) 80 mW, (**b**) 90 mW, (**c**) 100 mW.

**Figure 15 micromachines-14-01587-f015:**
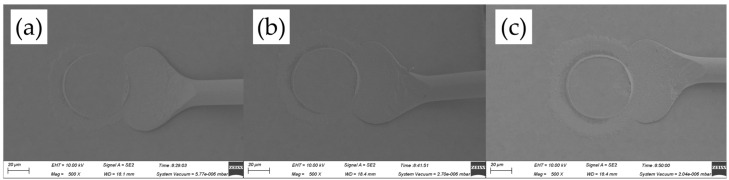
Wedge bond morphologies of AAPPCA wire for different ultrasonic powers at 95 gf: (**a**) 80 mW, (**b**) 90 mW, (**c**) 100 mW.

**Figure 16 micromachines-14-01587-f016:**
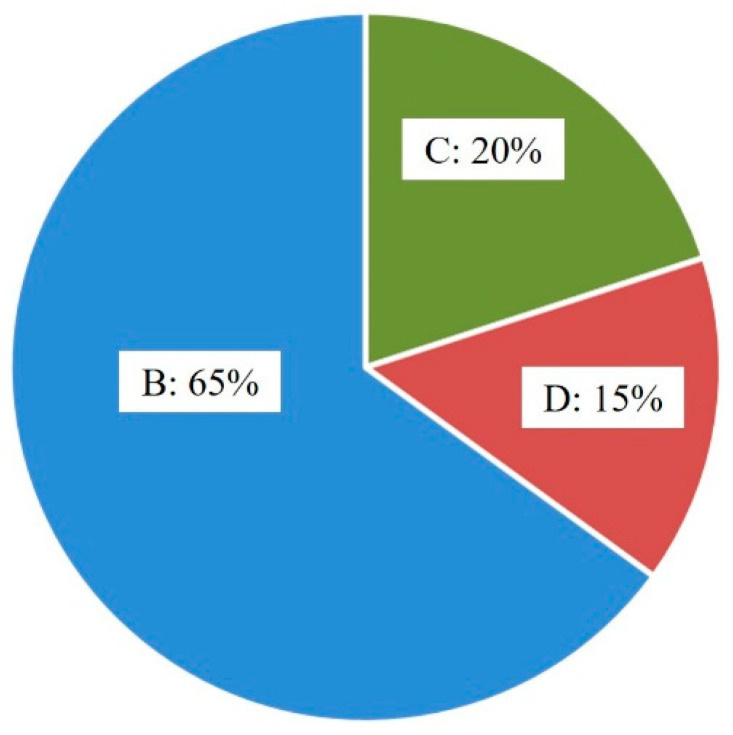
Percentage diagram of break locations after destructive pull test.

**Figure 17 micromachines-14-01587-f017:**
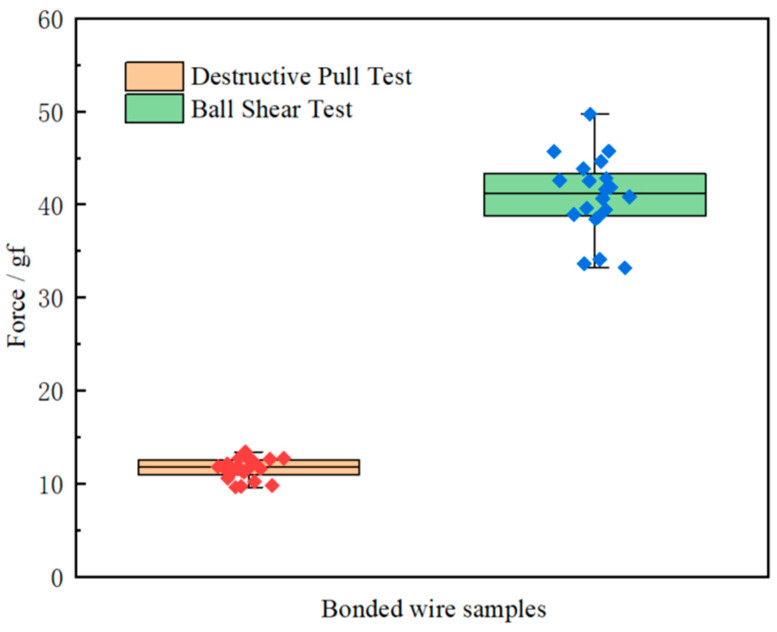
One-way analyses of pull and shear forces.

**Figure 18 micromachines-14-01587-f018:**
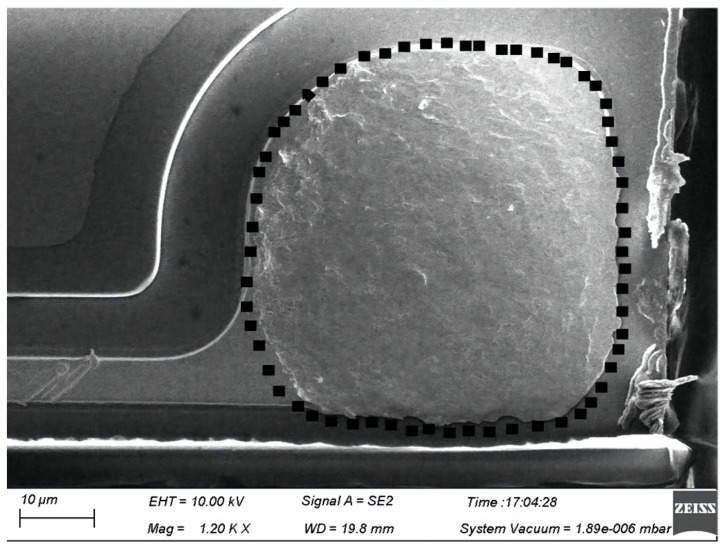
Typical ball bond morphology after ball shear test.

**Table 1 micromachines-14-01587-t001:** Process parameters for AAPPCA wire.

Free Air Ball		Ball Bond		Wedge Bond	
Spark Voltage/V	5000	Impact Force *F*_f_/gf	65	Bonding Force *F*_1_/gf	55
EFO Current/mA	20/25/30	Bonding Force *F*/gf	35/45/55	Ultrasonic Power *P*_1_/mW	60
EFO Time/μs	500/550/600/650/700/750/800	Ultrasonic Power *P*/mW	60/70/80	Bonding Time *t*_1_/ms	6
Tail Length/mm	0.15	Bonding Time *t*/ms	8	Bonding Force *F*_2_/gf	55/75/95
Bonding Temperature/°C	220			Ultrasonic Power *P*_2_/mW	80/90/100
				Bonding Time *t*_2_/ms	6

**Table 2 micromachines-14-01587-t002:** Destructive pull and ball shear tests results of bonded wire samples.

Samples	Destructive Pull Test		Ball Shear Test
	Break Location	Pull Force/gf	Shear Force/gf
1	B	12.5	40.6
2	D	10.2	42.5
3	C	12.7	39.6
4	B	9.7	38.4
5	B	11.5	34.1
6	D	13.4	41.6
7	B	12.1	45.7
8	C	11.4	38.9
9	B	10.6	33.2
10	B	9.8	42.6
11	B	12.8	43.8
12	C	11.6	39.4
13	D	11.9	41.8
14	B	12.6	42.8
15	B	9.6	49.7
16	B	11.8	33.6
17	C	12.7	40.8
18	B	12.4	38.6
19	B	11.7	45.7
20	B	11.2	44.6

## Data Availability

Not applicable.

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
