# Peer review of "Effects of Process Parameters on Bond Properties of Ag-2.35Au-0.7Pd-0.2Pt-0.1Cu Alloy Wire"

_micromachines, 2023, doi:10.3390/mi14081587_

Round 1

Reviewer 1 Report

In the reviewed article, the authors present the results of research on various properties on bond. These are interesting research results, but the question is: what practical applications do they have? Moreover literature review is insufficient. The authors should extend the literature review with the works of other researchers in the subject of research, taking into account the results by international authors.

Author Response

(1)The application of this study has been added into the introduction.

(2)More extensive research findings have been added to the literature review.

Reviewer 2 Report

The paper evaluates the effect of process parameters such as electrical flaming current and time, ultrasonic power, and bonding force on a wire of 25 um. The alloy used by the authors has some advantages compared with Au, Cu, and Ag wires as is highlighted by the authors in the introduction showing the importance of their work. Some optimal conditions were found by the authors evaluated on the free air ball generated, ball and wedge bond, in which a current of 25 um with 650 μs, and an ultrasonic 70 mW and 45 gf for ball bond was selected by the authors. I strongly recommend making improvements to the methodology presentation. The methodology should begin to present the material used, and the destructive pull and ball shear test should be explained in detail. The First paragraph in section 3.5 and figure 17 should be relocated to the methodology as well.

Something that I am concerned about that decreases the impact of your work is there is a similar work previously published by your research group (https://doi.org/10.3390/mi11080777), and despite the alloy used is different, the structure of the paper is practically the same. I think is necessary that you show your contribution based on that work. If you have improvement because of the alloy and the optimal conditions found. Otherwise, there is not any novelty or significance of content in your work. 

besides that, some minor corrections:  Tsai et al.  – include the reference number

Figures 1 – 2 - 6  and 12. Should be Located after you referenced it in the text

Line 81 - change N2 to N2

Line – 94 to 98 please, include a reference to reinforce this paragraph

Line 110 – change Ma to mA

Line 130 and 131 – the comma after mA, does not make sense – I think is appropriate to include an “and”. For instance in line 130: “Considering the occurrence of off-center FAB at the EFO time of 25 mA and 700 μs”.

Please – Put the number of Eq. 1 on line 160 – Although the 25 mA was optimal, in Fig 6 I believe could be interesting to include the data with other currents, and maybe include this important effect in the equation proposed.

Sentence line 163-165 Please, clarified that the model was calibrated for only 25 um. 

Author Response

(1) Thank you for your advice. The methodology presentation has been improved.

(2) As you mentioned, we have added Pt and Cu elements to the material, which was not present in our previous work. We hope that this study can provide technical support for the reliability research of Pt containing silver based bonding alloy wires

(3) The remaining minor changes have been made in the corresponding positions.

Round 2

Reviewer 2 Report

Congratulation on your contribution